# Who Is Afraid of Monkeypox? Analysis of Psychosocial Factors Associated with the First Reactions of Fear of Monkeypox in the Italian Population

**DOI:** 10.3390/bs13030235

**Published:** 2023-03-07

**Authors:** Filippo Maria Nimbi, Roberto Baiocco, Guido Giovanardi, Annalisa Tanzilli, Vittorio Lingiardi

**Affiliations:** 1Department of Dynamic and Clinical Psychology and Health Studies, Sapienza University of Rome, 00185 Rome, Italy; 2Department of Social and Developmental Psychology, Sapienza University of Rome, 00185 Rome, Italy

**Keywords:** monkeypox, fear, COVID-19, epistemic trust, mentalized affectivity, mental health

## Abstract

Background: A rising number of monkeypox cases have been detected in Europe and several Western nations. Evaluating the fear associated with monkeypox is crucial to determine the necessity for tailored education and prevention programs for specific populations. This study explores the psychological and social factors linked to the fear response to monkeypox. Methods: Nine self-report measures were completed by 333 participants (212 women, 110 men, and 11 individuals identifying as other genders) from the general Italian population, investigating different psychosocial variables. Results: The findings revealed that higher levels of monkeypox fear were linked to identifying as LGB+ or having close associations with the LGBTQI+ community, being single, having lower education levels, placing greater importance on religion, receiving more COVID-19 vaccine doses, having a lower current quality of life, and attributing increased impairment to the COVID-19 pandemic. Psychologically, higher levels of monkeypox fear were associated with higher levels of epistemic credulity, close-mindedness, anxiety, difficulty expressing emotions, and difficulty processing them. Conversely, lower levels of monkeypox fear were related to the belief that the media exaggerated the risks associated with monkeypox epidemics. A partial mediation model was presented and tested. Conclusions: Collecting and utilizing this data can help design targeted education and prevention programs to overcome the fear of monkeypox and promote healthier behaviors.

## 1. Introduction

Monkeypox is an infection caused by a virus belonging to the same family as smallpox (Poxviridae) but differs from smallpox in terms of its lower transmissibility and severity. Monkeypox was discovered and identified in 1958 in monk colonies [1]. In 1970, the initial instance of monkeypox affecting humans was documented in a nine-month-old infant in the Republic of the Congo [2]. Recently, beginning on 18 May 2022 [3], multiple occurrences of monkeypox have been identified in various Western nations. Public health services monitor and investigate the present outbreak and take measures to control it [4]. According to a recent meta-analysis [5], the current epidemic varies from previous outbreaks in terms of the affected age group (with 54.29% of cases occurring in individuals in their thirties), gender distribution (with most cases occurring in males), and mode of transmission. Specifically, the analysis found that the virus can be transmitted through close physical contact, including skin-to-skin contact and sexual activity, as well as through respiratory droplets and contaminated objects such as clothing, bedding, and sex toys [4]. Furthermore, the clinical presentation of the virus appears to be atypical, with anogenital lesions and rashes reported in 31.43% of cases [5].

There are many uncertainties around the recent outbreak of monkeypox in Europe [6], and they should be managed wisely through risk communication and community engagement strategies considering what is still unknown about monkeypox. The lack of knowledge regarding monkeypox mainly refers to undetected transmission, an unusual presentation of the symptoms, and additional ways of transmission [7]. There is a high risk of stigmatizing certain populations, such as African individuals and men who have sex with men (MSM), in the current epidemic. In fact, “Social Science in Humanitarian Action (SSHAP)” has released a paper [6] discussing these risks and suggesting ways to improve communication regarding this emergency. When developing communication strategies, it is important to take into account lessons learned from past epidemics, such as HIV/AIDS and COVID-19. In both instances, the dissemination of certain information has fueled fear and discrimination towards specific groups (such as MSM, transgender individuals, sex workers, individuals who inject drugs, and Chinese individuals), and unfortunately, the stigma associated with these groups still persists today [6,8,9].

To date, we do not know what the social reaction to the spread of monkeypox may be. One of the main psychological aspects usually related to pathogens’ responses is fear. Fear is an emotion that has significant effects on physiology and behavior. The term “fear” is used to describe the feeling that arises when an impending threat to survival is experienced by the individual [10]. Research highlighted the biological, evolutionary, and social basis of fear, both referable to individual and collective explanations. From a neurobiological perspective, many studies have addressed the fear among different species [10,11,12,13,14]. Fear is generally processed in independent neural routes that involve the amygdala and downstream hypothalamic and brainstem circuits [10]. When near or imminent, threats activate monitoring systems and defense reactions that promote self-preservation. The amygdala and bed nucleus seem to be involved in the process of threat prediction and monitoring. In contrast, the orbitofrontal cortex seems to be involved in the emission of safety signals that contrast the fear [13].

From a psychological perspective, although fear and anxiety are similar and often confused in the literature, the first is recognized as a generally adaptive state of apprehension that quickly begins and dissipates once the threat is removed. Anxiety is aroused by less specific and predictable threats, or by physically or psychologically more distant threats. Anxiety is thus a more enduring state of apprehension [12]. In this sense, a study of emotional reactions to a new possible threat such as monkeypox should consider both factors.

From a more social perspective, uncertainty related to pathogens may lead people to a binary state of mind, where external information may affect their mental health and affective (and behavioral) reactions both positively and negatively. In this sense, the recent literature on COVID-19’s first reactions may be helpful to better understand the role of fear and anxiety in exposure to pathogenic risks. The outbreak of COVID-19 in China has created uncertainty among the public, and the official media have intensified this sense of uncertainty due to the late reporting of the real epidemic situation. At the same time, unofficial media were quicker to spread news and fill the gaps, although often contradicting the official ones. This created doubts about the authenticity of the information provided by the media, negatively influencing public sense of trust, vulnerability, mental health, and health behavior [15]. Other studies during the national lockdowns in 2020 reported high worry and fear about COVID-19 [16,17]. In this sense, the heightened media coverage reporting the daily number of infections and deaths that has characterized the pandemic’s first months showed profound psychological effects worldwide in responses of fear, anxiety, and isolation [18,19].

Assessing monkeypox-related fear is central to evaluating the need for education and prevention programs tailored to specified populations based on sociodemographic variables and other factors [6,19]. For example, a recent meta-analysis on COVID-19 [20] shows that women reported higher levels of fear than men, with Asian people and hospital staff being the most scared populations. In the US, fear was higher in regions with the highest reported COVID-19 cases and socially vulnerable respondents such as women and Chinese and Hispanic people [17]. Moreover, they also showed a high association between fear and mental health sufferance due to anxiety and depressive symptoms.

The study of protective factors from fear is also central. Coelho et al. [21] showed that some factors were protective in dealing with the fear and emotional consequences of COVID-19, such as tolerance of the unknown, lower levels/sensitivity to anxiety, resistance to social isolation, low levels of sensitivity to disgust, receipt of financial support, use of caution concerning media coverage, and higher levels of efficacy. 

In this sense, the study of emotional reactions and associated psychosocial factors to monkeypox appears to be urgently needed, also exploring the possible associations of variables related to the reliability of news and emotional regulation such as epistemic trust and mentalized affectivity, respectively. Epistemic trust [22] refers to an individual’s willingness to consider new knowledge as reliable and relevant, thus incorporating it into their personal life. Conversely, epistemic distrust is characterized by rigid thinking patterns and difficulty learning from the social environment. Mentalized affectivity [23] is the cognitive and emotional ability to comprehend our own and others’ thoughts and feelings. Affective regulation of emotions relies on mentalization, which involves identifying, processing, and expressing emotions, while also re-evaluating their meaning. The theoretical framework of this study is rooted in mentalization [22], which is a developmental process dependent on secure attachment relationships that accurately reflect one’s subjective experiences, allowing for the establishment of secondary representations of these experiences [24]. Previous research has shown links between attachment security, mentalization, and the ability to generate epistemic trust [22]. The current hypothesis is that stable traits (such as mentalization, epistemic trust, closed-mindedness, and religiosity) may impact health behavior and social attitudes, particularly when fear is triggered by uncertainty and the unknown.

### Aims

Given the recent emergence of the monkeypox epidemic in Europe and Western countries (May 2022), there is currently a lack of studies examining individuals’ emotional and psychological reactions to the outbreak. This study seeks to address this gap by exploring potential predictors of fear reactions to monkeypox using a set of psychosocial variables that have been previously studied in the context of epidemics such as HIV and COVID-19.

Specifically, the study has two primary objectives. The first objective is to investigate the impact of various factors on fear reactions to monkeypox, including (1) sociodemographic variables, (2) attitudes and knowledge regarding monkeypox and experiences during the COVID-19 pandemic, (3) psychological variables such as epistemic trust, need for closeness, psychopathological traits, and mentalized affectivity, and (4) attitudes towards the LGBTQI+ community and sexuality.

The second objective of the study is to identify the best predictors of fear of monkeypox from a biopsychosocial perspective and test a path diagram that examines the relationships between the main variables related to fear. The study aims to discuss the possible implications of these findings for health and prevention policies.

## 2. Materials and Methods

### 2.1. Procedures

A total of 342 individuals from the general population in Italy participated in this study, consisting of 218 women, 112 men, and 12 individuals identifying as other genders. Participants were recruited through a snowball technique and paid advertisements on institutional websites and social media platforms such as Facebook, Instagram, and LinkedIn. The online survey was conducted through the Google Forms platform and responses were collected in June 2022. Participants were required to provide informed consent before taking part in the survey, and the questionnaire was anonymous with no compensation provided. This study was approved by the institutional ethics committee of the Department of Dynamic and Clinical Psychology and Health Studies at Sapienza University of Rome, Italy, on 13 June 2022 [Protocol number 0000994 UOR: SI000092—Classified VII/15]. The inclusion criteria for participation in the study were being 18 years or older and fluent in Italian. Nine responses (2.63%) were excluded due to duplication, falsification, or incomplete records, resulting in a final sample size of 333 participants (212 women, 110 men, and 11 individuals identifying as other genders). Table 1 provides an overview of the sociodemographic characteristics of the participants.

### 2.2. Participants

The inclusion criteria were as follows: (1) age 18 or over; (2) being fluent in Italian. Nine responses (2.63%) were excluded from the present study because they represented duplicated, falsified, or incomplete records. The final group resulted in 333 participants (212 women, 110 men, and 11 other genders). Sociodemographic data of the participants are presented in Table 1. 

The participants in the study had a mean age of 31.71, with most being under 40 years old. The majority of participants were assigned female at birth and identified as female, and most participants reported being heterosexual, unmarried, and in a relationship, with a small percentage identifying as polyamorous. The group had medium to medium-high levels of education, with about half of the participants being employed. Almost all of the participants were white Caucasian, lived in a large city, and had medium to medium-low socioeconomic status. In terms of political and religious orientation, nearly half identified as center-left, while the majority did not identify with any religion. However, most participants reported having had a religious upbringing. It is worth noting that the same group of participants was also used in another study that examined attitudes towards monkeypox fake news [25].

### 2.3. Measures

For this study, participants were assessed with nine self-report measures to explore various psychosocial variables. The administration time for these measures was approximately 25 min.

Sociodemographic questionnaire—Participants were asked to complete a brief sociodemographic form to gather general information such as age, gender, sexual orientation, marital and relational status, education level, work status, socioeconomic status, ethnicity, area of residence, religious education (coded from “0. Not at all” to “4. Very much”), religiosity (i.e., “importance given to religion in their life”, coded from “0. Not at all” to “4. Very much”), and political orientation (coded from “0. Extreme left” to “4. Extreme right”).

Monkeypox knowledge—An ad hoc questionnaire consisting of five items was created based on information published on 23 May 2022 in the ECDC [4] report “Rapid risk assessment. Monkeypox multi-country outbreak” to evaluate the level of knowledge about monkeypox. The questionnaire consisted of questions such as “Monkeypox is caused by: [...]”, “Monkeypox is transmitted: [...]”, “One can be infected with monkeypox through: [...]”, “The incubation period of monkeypox is: [...]”, and “The main symptom(s) of monkeypox is/are: [...]”. For each question, five alternative answers were presented, of which only one was correct. Correct answers were counted with one and summed to create the monkeypox knowledge scale.

Attitudes towards monkeypox—Five questions were created to evaluate attitudes towards monkeypox. Participants were asked to respond on a 6-point Likert scale ranging from 0 Not at all to 5 Very much. “How scared do you feel about monkeypox?”; “How much do you feel at risk of contracting monkeypox?”; “How far do you think monkeypox can spread to the point of becoming a pandemic (such as COVID-19)?”; “How much do you think monkeypox is a hoax or fake news?”; “How much do you think the media are amplifying or exaggerating the danger of monkeypox?”. From a preliminary factor analysis, questions on fear, risk of contagion, and fear of monkeypox becoming a pandemic merged into the “Fear of monkeypox” domain, this study’s central variable of interest.

The COVID-19 quality of life (QoL) was also measured using an ad hoc questionnaire. This questionnaire measured the perception of participants’ current QoL and self-attributed QoL changes due to the COVID-19 pandemic. The questionnaire included single items about having contracted COVID-19 from the beginning of the pandemic, the number of vaccine doses taken, and proximity to no-vax (anti-vaccine) positions and ideals [26]. Exploratory factor analysis demonstrated good reliability of the measure of perceived QoL and worsening attributed to the COVID-19 pandemic.

Epistemic trust, mistrust, and credulity questionnaire (ETMCQ) [27,28] is a 15-item self-report questionnaire that assessed participants’ epistemic trust, mistrust, and credulity towards communication or communicated knowledge. Higher scores indicated a higher presence of the relative trait for each factor. The validity of the ETMCQ domains demonstrated good internal consistency. The Cronbach’s alpha values for this measure in the current study ranged from 0.86 (credulity) to 0.9 (trust).

Need for closure scale—short form (NCS) [29] measures five dimensions related to the need for cognitive closure: order, predictability, decisiveness, ambiguity, and close-mindedness. It is the short form of the need for cognitive closure [30,31]. These dimensions are separate but related aspects within the theoretical framework for the cognitive-motivational aspects of decision-making. The need for closure is characterized by two tendencies: urgency and permanence. Urgency refers to the predisposition to “seize” closure quickly, while permanence refers to the desire to maintain or “freeze” closure. Individuals with a high need for closure prefer order and structure in their lives, abhorring chaos and disorder, and have a closed mindset, being unwilling to challenge their knowledge with alternative opinions or inconsistent evidence. The short form is valid and has good psychometric properties, with higher scores indicating a higher presence of the relative trait for each factor. The Cronbach’s alpha values for this measure in the current study ranged from 0.78 (decisiveness) to 0.89 (order).

Brief symptom inventory 18 (BSI-18) [32] is a widely used short form of the symptom checklist 90-revised (SCL90-R). It contains three six-item scales related to somatization, depression, and anxiety, as well as a global severity index (GSI). Higher scores indicate a more significant presence of symptoms reported in that domain. The BSI-18 has satisfactory psychometric properties and has demonstrated good results in previous studies. The Cronbach’s alpha values for this measure in the current study ranged from 0.91 (somatization) to 0.93 (anxiety).

Brief-mentalized affectivity scale (B-MAS) [33,34]*—*The B-MAS is a self-report questionnaire composed of 12 items that assess 3 aspects of mentalized affectivity involved in the regulation of emotions. These aspects include identifying emotions, processing emotions, and expressing emotions. The ability to mentalize affectivity is indicated by higher scores on the B-MAS. In the current study, Cronbach’s alpha values for this measure ranged from 0.88 to 0.91.

LGBTQI+ and sexual attitudes*—*To evaluate attitudes towards LGBTQI+ people and sexuality, various measures were utilized in the study. One measure involved a visual item in which respondents placed themselves at different distances from the LGBTQI+ community (see Appendix B). Another measure included ten items from the “Attitudes toward Lesbians” and “Attitudes toward Gay Men” scales [35], with higher scores indicating worse attitudes. The Cronbach’s alpha values for this measure in the current study ranged from 0.86 (lesbians) to 0.89 (gay men). Additionally, eight items from the sex-positive attitudes scale [36] were used to assess attitudes towards sexual mores, with higher scores indicating more rigid and moralistic positions.

Marlowe-Crowne social desirability scale—short form (MCSDS—SF) [37] was used as a covariate in the analysis to control for social desirability bias. The MCSDS—SF is a 13-item measure that assesses the tendency to respond in a socially desirable way, with higher scores indicating a stronger tendency to do so. In the current study, the Cronbach’s alpha value for this measure was 0.91.

### 2.4. Statistical Analysis

Table 2 presented the primary variables of interest for the study: Hierarchical multiple regression analyses using the enter method were conducted for each area assessed to identify significant predictors of monkeypox fearful attitude. The sub-scales of each questionnaire were used as independent variables, and “Fear of monkeypox” was the dependent factor. The regression analyses were conducted in four stages, following methodological suggestions of Petrocelli [38] and Lewis [39]. The stages were as follows: (1) demographics, socioeconomic variables, and political and religious orientation; (2) monkeypox attitudes and COVID-19 QoL and attitudes; (3) epistemic trust, need for closure, psychopathology, and mentalized affectivity; (4) LGBTQI+ attitudes and sexual attitudes.

A final hierarchical multiple regression analysis was performed, including the significant variables from the previous regressions, to identify the best predictors of “Fear of monkeypox” (enter method). The MCSDS—SF was used as a covariate in all hierarchical multiple regression analyses to minimize the effects of social desirability bias. The results of the MCSDS—SF are only presented for the final regression for simplicity.

A theory-driven path analysis model was also developed, with variables referring to more stable traits hypothesized to influence behavior and attitude variables. Fit indices, including chi2, goodness of fit (GFI), normed fit index (NFI), comparative fit index (CFI), and root means square error of approximation (RMSEA), were used to assess the model fit. IBM SPSS v. 27.0 (SPSS Inc., Chicago, IL, USA) and SPSS AMOS were used to perform the statistical analyses.

## 3. Results

Table 2 presents descriptive data on attitudes and knowledge regarding monkeypox and COVID-19. The group’s monkeypox knowledge appears to be balanced between those who answered the questions correctly and those who did not. The most common sources of information for the group were newspapers and TV news, followed by the Internet and social media. In terms of attitudes toward monkeypox, most participants reported feeling little fear about the virus, considering the risk of contracting it to be low, and not fearing a potential pandemic such as COVID-19. More than 60% did not consider monkeypox to be fake news, but most believed that the media exaggerates the danger of the virus. In terms of adherence to COVID-19 health policies, over half of the group reported having contracted COVID-19 at least once, and around 75% had received three doses of the anti-COVID vaccine, while approximately 10% had not received any doses. Nearly a quarter of the group expressed some level of agreement with anti-vaccine positions during the pandemic. 

A minimum of 140 participants was calculated a priori to ensure adequate statistical power (0.8) to run the following analyses, which included 15 predictors and a minimum effect size of 0.15. The actual sample size for the hierarchical multiple regression analyses was 333 participants, resulting in a post-hoc observed statistical power of 0.99.

To identify the best predictors of monkeypox fear, several classes of multiple hierarchical regression analyses were performed (enter method, Table 3, Table 4, Table 5 and Table 6), with social desirability as a covariate and the domains of each questionnaire as independent variables. Predictors that emerged as significant included sexual orientation (heterosexual vs. LGB+), relationship status, education level, religiousness, media amplification, perceived QoL, worsening QoL attributed to COVID-19, number of COVID-19 vaccine doses, epistemic credulity, close-mindedness, anxiety, processing emotions, expressing emotions, and closeness to the LGBTQI+ community. Higher levels of monkeypox fear were associated with being non-heterosexual and/or close to the LGBTQI+ community, being single, having lower education levels, giving more importance to religion, having received more COVID-19 vaccine doses, reporting lower levels of current QoL, and increased impairment attributed to the pandemic. On the psychological side, higher levels of monkeypox fear were associated with higher levels of epistemic credulity, close-mindedness, anxiety, ability to express emotions, and difficulties processing emotions. In contrast, lower levels of monkeypox fear were associated with a belief that the media exaggerates the risks of monkeypox epidemics.

To determine the strongest predictor of monkeypox fear, a final hierarchical multiple regression analysis was performed. In this analysis, social desirability was used as a covariate (Table 7, Step 1), and the factors that were found to be significant in previous analyses were used as predictors (Table 7, Step 2). The results showed that the model was significant and explained 32.4% of the variance in monkeypox fear (F_(15,317)_ = 10.13, *p* < 0.001, ΔR^2^ = 0.32). The predictor’s religiousness, media amplification, number of COVID-19 vaccine doses administered, epistemic credulity, anxiety, and expressing emotions significantly emerged as significant predictors of monkeypox fear. In contrast, epistemic credulity appeared to be the strongest one.

The authors aimed to test some of the variables identified in a partial mediation model using standardized equation modeling. Two variables, “expressing” and “social desirability,” were excluded from the analysis due to their theoretical proximity and collinearity, which could lead to the misinterpretation of the model. Figure 1 shows the path diagram constructed for the remaining variables, taking into account not only their direct impact on monkeypox fear but also their interactions. The main variables selected were epistemic credulity and religiosity, while anxiety and the number of vaccine doses were identified as endogenous variables that depend on credulity and religiosity. Media amplification of monkeypox risk was considered an exogenous variable, as it is independent of the other predictors.

Considering having 333 individuals as a factor hindering the power of chi^2^-based analyses, the model showed a satisfactory fit to data (chi^2^ = 7.239; df = 6; *p* = 0.299; GFI = 0.993; NFI = 0.959; CFI = 0.992; RMSEA = 0.025 [95% CI 0.000–0.079]). All the endogenous paths were found to be significant (Figure 1). The standardized total effects of credulity and religiousness on monkeypox fear were large and medium (credulity = −0.314, *p* < 0.001; religiousness = 0.207, *p* < 0.001). About mediations, credulity was significantly indirectly connected with monkeypox fear through anxiety (indirect effect = 0.061, *p* < 0.01). Religiousness was not significantly connected with monkeypox fear through anxiety and number of COVID-19 vaccine doses. This model explained the 25.4% variance in monkeypox fear.

## 4. Discussion

The study aimed to recruit a diverse group of participants in terms of sociodemographic characteristics, and participants showed a wide range of knowledge about monkeypox based on the ECDC report [4]. Most participants did not express fear of monkeypox, and only a few were concerned about contracting the virus or the possibility of a pandemic such as COVID-19. As of 1 July 2022, there were 192 confirmed cases of monkeypox in Italy, with a higher incidence in men, primarily those who had traveled abroad [40]. The report did not provide information about the sexual orientation of the infected men or the countries they visited. However, the male predominance of the cases may contribute to the perceived threat level among the general population. Initially, some people may have attributed the epidemic to men who have sex with men (MSM) traveling to LGBTQI+ international events [41], similar to the initial reactions towards HIV more than 40 years ago.

In line with the study’s objectives, an attempt was made to reach a group of participants varied in sociodemographic characteristics. Regarding information and knowledge about monkeypox based on the ECDC report [4], a wide variance emerged among participants showing how information has been differently acknowledged. Most participants did not seem afraid of monkeypox; few reported feeling at risk of becoming infected and worried that it could become a pandemic such as COVID-19. At the time of closing data collection (1 July 2022), 192 cases of monkeypox were confirmed in Italy [40], with a primary relevance of men (n = 190) with 68 cases related to travel abroad. The report does not mention the sexual orientation of the men infected or the countries where they traveled. However, the male prevalence may influence the perceived dangerousness for the general population, which at first tended to relegate this epidemic to MSM coming from LGBTQI+ international events [41] as it was for HIV more than 40 years ago. 

In line with the first objective, some interesting profiles emerged: participants who reported higher scores on the fear scale were more likely to be LGB+, single, less educated, and higher religiosity. The presence of sexual orientation as a predictor can be due to the fact that LGB+ participants may feel more at risk for monkeypox, in line with the early news shared by media on the monkeypox outbreak which spoke of an epicenter during popular LGBTQI+ events [6]. In addition, other data support how the risk of contagion within the MSM population compared to other groups could be higher at the moment [5]. In this sense, health communication and prevention policies should address the MSM population and the LGBTQI+ community sensitively and inclusively, avoiding the errors of communication that can trigger fear and anxiety states, activating different levels of intersectional stigma [6,42,43,44]. At the same time, having greater proximity to the LGBTQI+ community seems to indicate more fear of monkeypox. As such, communication about monkeypox should not focus exclusively on MSM but should involve the entire population considering a broader range of potential contacts of people affected by monkeypox [6]. It is emphasized that with an exclusive focus on MSM, other populations may not be seen as vulnerable, minimizing risks to other high-risk groups, including untreated HIV/AIDS patients, sex workers, and others. In fact, prevention messages should also include all known transmission routes and not be limited to sexual transmission. Furthermore, while people closer to the LGBTQI+ community may be more sensitive to the risk and are more likely to join screening and prevention campaigns, messages aimed at the entire population should raise awareness among those who feel less at risk. For example, it is highly suggested at a national level (in Italy as well as abroad) to plan an information campaign directed at the entire population that would allow easier access to preventive devices such as vaccines and healthcare. The specific topics to be addressed could be the fear of contagion, the symptoms, and the possible routes of transmission. In this sense, combining monkeypox prevention with campaigns that could address stigma towards MSM and sexuality would potentially have the dual benefit of improving sexual health levels and decreasing discriminatory attitudes towards the LGBTQI+ community under a sex-positive approach [42]. It should also be noted that attitudes toward gays and lesbians did not appear to be correlated with fear of monkeypox in this study, suggesting that associations between stigma related to sexual expression and monkeypox are not yet detectable at this early stage of the epidemic. However, it is not excluded that they may be detected by other means or at different stages of epidemic evolution.

The increased presence of fear in single participants could be related to the fact that the idea of sexual behaviors as the main route of contagion has rapidly spread, bringing monkeypox into the imagination of sexually transmitted infections [45]. This would perhaps allow people in a relationship (especially a monogamous one) to feel less at risk for possible contagion. The discourse on religiosity is very complex as it includes how individuals experience their faith. Participants who claimed to confess a religion were mainly Catholic Christians. One possible explanation of the connection between religiosity and fear of monkeypox could be that the narrative of monkeypox as a pathology related to sexual promiscuity among MSM could be seen as a sign of sin and divine punishment. In this sense, prejudice could fuel the fear of corruption and contagion [46]. This hypothesis would need to be verified by further studies. However, an inclusive narrative that seeks to dismantle stereotypes and stigmas about different forms of sexuality could positively affect fear and screening deconstructing moral, religious, and cultural aspects related to the way we interpret sexuality [42].

Unlike what has been reported in the literature regarding other diseases, such as in references [47,48], this study found that education had an inverse relationship with fear. However, it was not found to have a direct impact on specific knowledge related to monkeypox. A lower level of education was found to be associated with less specific knowledge about the disease, and less confidence in managing the health risk [49]. The study also revealed that epistemic credulity played a significant role in predicting fear. As previously posited by the authors of the ETMCQ [27], credulity refers to a lack of discernment and vigilance towards information, indicating an overall confusion about one’s stance and resulting in a susceptibility to misinformation and potential exploitation. This result is interesting when crossed with the idea shared by many participants that the media are amplifying the risk and danger of monkeypox, which seems to trace the epistemic stance of mistrust [27,49,50,51]. Campbell et al. [27] discussed how a negative relationship between credulity and general self-efficacy could be translated into poor enactment of healthy behaviors towards a health danger. In this sense, the emotion of fear could become an element that hinders monkeypox risk management rather than a factor that motivates the implementation of preventive and screening behaviors. 

Consistent with the discussion on credulity, the study found that individuals who are more close-minded are more likely to experience higher levels of fear. Close-mindedness refers to the tendency to seek secure closure, which can result in a reluctance to have one’s beliefs challenged by opposing views or inconsistent evidence [30]. This cognitive need for closure may simplify the response to a perceived threat, leading to stereotypical and stigmatizing thoughts and behaviors that exacerbate fear. A study on COVID-19 [52] also demonstrated that when the causes of the epidemic are unclear, close-minded attitudes and rumors can increase, leading to fear, stigma, and negative behaviors such as violence and suicide during the early stages of the pandemic.

For COVID-19 and perceived QoL, participants reporting worse current QoL and recognizing a worsening of their condition due to the COVID-19 pandemic reported a higher degree of fear toward monkeypox. Moreover, fear seems to be higher in those who have completed the vaccination cycle, which in Italy includes three doses (the fourth dose is recommended at the moment of writing for elderly and frail individuals). This may represent a consequence of the long-term stress effects of COVID-19 since we know very little despite the large number of studies that have been produced on COVID-19 psychological effects. A meta-analysis [53] found that worse levels of stress and mental health due to the pandemic period increased emotional reactions to stress and the presence of psychopathological symptoms. As such, an increased level of perceived stress may be more easily translated into a response of despondency, anger, and fear in the face of a possible new attack, especially in those individuals with more precarious QoL. In fact, also in the current study, the presence of anxiety symptoms was associated with a higher level of monkeypox fear. This result is not surprising since anxiety and fear reactions are often bidirectionally linked. Studies performed on COVID-19 [17,19,54] also confirmed this strong association where higher levels of fear were related to higher rates of anxiety symptoms. It is important to reiterate the emphasis on mental health in preventing and managing infectious diseases [55]. In addition, one can speculate how adherence to mass vaccination for COVID-19 has mainly affected more fearful and anxious people, who tend to react similarly to monkeypox [56]. There is a need to reassure and care for people rather than convey messages that may frighten them into increasing levels of trust and engagement with the health care system [6]. 

Regarding mentalized affectivity, on the one hand, an impairment of the emotional processing function appears to limit the identification and reporting of monkeypox fear. The fear of an unknown and unseen threat, especially in a context already overwhelmed by fear and exhaustion from dealing with another epidemic, can make emotional processing challenging. This can lead to emotional outbursts in individuals who have less developed emotional regulation skills [57]. Conversely, having a greater ability to express emotions inwardly and outwardly allows for greater ease in expressing fear. This is primarily seen in therapeutic settings, where this ability is extensively cultivated [33,57]. Research shows [58] that good mental capacity is a protective factor for psychological health, which can be improved with proper education and training programs. During the COVID-19 pandemic outbreak, good mental affective mentalization skills significantly reduced mental health disorders. 

In line with the second objective of this study, the final hierarchical multiple regression highlights among all the variables discussed so far, some that seem to play a predominant role, such as epistemic credulity, anxiety, religiousness, and the idea that media amplify the news about monkeypox emerged as the best predictors of monkeypox fear. In addition to these, but with relatively lower statistical power, the number of doses of COVID-19 vaccine administered and emotional expression were also significant predictors. 

The presented mediation model aims to illustrate how certain structural traits, such as epistemic credulity and religiosity, may have an indirect impact on fear levels through anxiety responses and health-related behaviors, specifically the number of COVID-19 vaccine doses administered, which are linked to a prior health emergency. There should be recognition of the role of complexity linking these psychosocial variables, in which the fear response seems more common in those profiles characterized by epistemic credulity, presence of anxiety symptoms, the greater expressive ability of emotions, more doses of vaccine administered, and a higher level of religiosity. Authors also point out how distrust of the news given in the media can drive away a sense of fear and, perhaps, interest and awareness of the monkeypox outbreak [6]. Understanding this information is important for developing prevention campaigns and exploring novel constructs such as epistemic trust and mentalized affectivity [33,57]. It is crucial to increase awareness and take steps to reduce epistemic credulity and its effects to improve adaptation to interpersonal environments and increase the effectiveness of health strategies, ranging from prevention campaigns to psychotherapeutic interventions. This argument applies not only to monkeypox but also to health communication in general.

However, there are some limitations to the present study that should be addressed. Firstly, participants were selected using a “snowball” technique and social media advertisement, making it impossible to generalize the results to the entire Italian population despite the variety of participants involved. Secondly, some of the measures used in the study were ad hoc created to measure reactions and attitudes towards monkeypox and COVID-19. Although preliminary psychometric analyses were conducted, further studies are needed to investigate the validity and reliability of these constructs. Then, the study was based on self-report questionnaires, which may be easily falsified by respondents. To mitigate this bias, the study included social desirability measures as a control variable. Lastly, in the present study the distinction between a more “reasonable” fear related to an actual risk of exposure to the virus (e.g., people who have a high number of sexual partners, people who participate more frequently in mass events) and a more irrational fear (phobia) due to a pathophobic propensity of the person was not investigated. This element will be the subject of an upcoming qualitative study.

In future research, it would be valuable to examine cross-cultural differences in reactions to monkeypox. Additionally, it would be important to conduct a survey specifically within the LGBTQI+ community, particularly among MSM, to better understand how reactions and associated distress may evolve over time [5]. Furthermore, it would be beneficial to explore other significant psychological constructs such as personality traits, attachment styles, and defense mechanisms.

## 5. Conclusions

The current study evaluated the psychosocial factors associated with monkeypox fear in Italy. Several factors were identified as significant, including epistemic credulity, anxiety, religiosity, and the perception that the media disproportionately emphasize monkeypox news. These findings have important implications for the development of effective health strategies aimed at prevention and psychological support. Although the data presented in this study are preliminary, they can be used to inform practitioners, policy makers, and decision-makers in their efforts to manage and control the outbreak. Some preliminary risk factors can be identified (e.g., being single, being part of or being close to the LGBTQI+ community, having a low level of education), as well as some elements that can compromise health communication (distrust of the media). To cope more broadly, a massive approach covering several educational fronts is suggested in Italy: on the one hand, the promotion of more internal resources related to the ability to discriminate media news and to rely on safer information (scientific and media literacy) and on the other hand, a massive work on sexual health, equal rights and the safeguarding of the psychological health of minorities is considered urgent and necessary by the authors. This would make it possible to address the health threat posed by not only monkeypox but also future health emergencies that risk social fallout such as stigma and inequality.

As stated by Pakpour and Griffiths [19] for COVID-19, without knowing the level of fear in various contexts, between the different groups, and the different psychological factors at play, it is not very easy to tell if education and prevention programs are needed and, if they are, which targets to involve. Collecting and applying such data could be used to design targeted education and prevention programs to help overcome the fear of monkeypox and support such individuals engage in healthier behaviors.

Despite the current uncertainty surrounding monkeypox outbreaks, it is important to utilize resources and lessons learned from past infectious disease outbreaks [6]. Now is a crucial time to put these lessons into action and ensure that risk communication is based on evidence and does not perpetuate stigma or discrimination. It is important to expand considerations of those at risk and focus on collaborative communities and institutions with a biopsychosocial perspective to improve quality of life [42].

## Figures and Tables

**Figure 1 behavsci-13-00235-f001:**
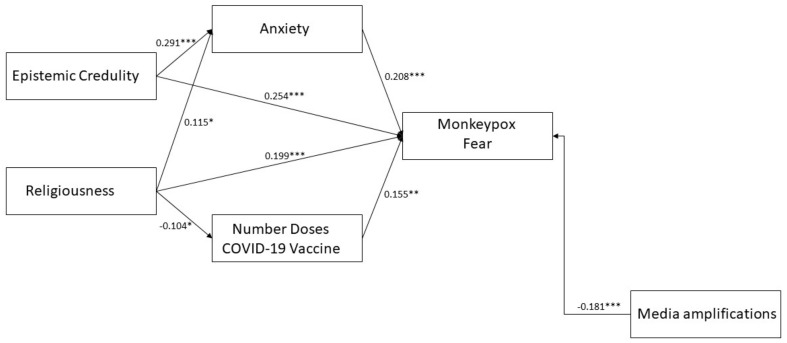
Path diagram partial mediation model of epistemic credulity and religiousness on monkeypox fear. Note: * = *p* < 0.05; ** = *p* < 0.01; *** = *p* < 0.001. Observed variables and residual errors are omitted for simplicity. A significant covariance path (r = −0.407 *p* < 0.001) was draw between errors of Media amplifications and Number Doses COVID-19 Vaccine to increase the model fit.

**Table 1 behavsci-13-00235-t001:** Sociodemographic variables description.

Variables		Participants (n = 333)
		**M ± ds (min-max)**
Age		31.71 ± 11.14 (18-71) Q_3_–Q_1_: 22–40
		**n (%)**
Sex assign at birth	Female	217 (65.17)
	Male	116 (34.83)
Gender	Female	212 (63.66)
	Male	110 (33.03)
	Transgender	1 (0.3)
	Non-binary spectrum	4 (1.2)
	Currently exploring gender identity	6 (1.8)
Sexual Orientation	Heterosexual	237 (71.17)
	Bisexual	31 (9.31)
	Homosexual	50 (15.02)
	Pansexual	11 (3.3)
	Asexual	4 (1.2)
Marital Status	Unmarried	269 (80.78)
	Married	53 (15.92)
	Separated	10 (3)
	Widowed	1 (0.3)
Relational Status	Single	162 (48.65)
	Couple	160 (48.05)
	Polyamory	11 (3.3)
Education Level	Middle School	16 (4.8)
	High School	140 (42.04)
	University	131 (39.34)
	PhD and Postgrads courses	46 (13.81)
Work Status	Unemployed	32 (9.61)
	Employed	179 (53.75)
	Student	120 (36.04)
	Retired	2 (0.6)

**Table 2 behavsci-13-00235-t002:** Monkeypox and COVID-19 variables description.

Variables		Participants (n = 333)
		**M ± ds (min-max)**
Monkeypox knowledge		3.04 ± 1.1 (0–5) Q_3_–Q_1_: 2–4
Fear about Monkeypox (Total Score)		3.54 ± 2.42 (0–15) Q_3_–Q_1_: 2–5
		**n (%)**
Where did you mainly acquire your knowledge about monkeypox?	From the news and newspapers	109 (32.73)
	From social media	80 (24.02)
	From friends, relatives, and colleagues	18 (5.41)
	I have searched info on internet	86 (25.83)
	Books and other scientific fonts	20 (6.01)
	I have not received any information about monkeypox	20 (6.01)
How scared do you feel about monkeypox?	Not at all	82 (24.62)
	Little	141 (42.34)
	Moderately	75 (22.52)
	Quite	25 (7.51)
	Very	6 (1.8)
	Very much	4 (1.2)
How much do you feel at risk of contracting monkeypox?	Not at all	92 (27.63)
	Little	164 (49.25)
	Moderately	61 (18.31)
	Quite	9 (2.7)
	Very	3 (0.9)
	Very much	4 (1.2)
How far do you think monkeypox could spread so far as to become a pandemic (as happened with COVID-19)?	Not at all	70 (21.02)
	Little	151 (45.35)
	Moderately	80 (24.02)
	Quite	20 (6.01)
	Very	7 (2.1)
	Very much	5 (1.5)
How much do you think monkeypox is a hoax or fake news?	Not at all	206 (61.86)
	Little	75 (22.52)
	Moderately	23 (6.91)
	Quite	10 (3)
	Very	7 (2.1)
	Very much	12 (3.6)
How much do you think the media are amplifying or exaggerating the danger of monkeypox?	Not at all	39 (11.71)
	Little	68 (20.42)
	Moderately	91 (27.33)
	Quite	72 (21.62)
	Very	28 (8.4)
	Very much	35 (10.51)
Number of times COVID-19 has been contracted	Never	151 (45.34)
	Once	158 (47.45)
	Twice	24 (7.21)
Number of doses of anti-COVID-19 vaccine administered	Zero	34 (10.21)
	One	2 (0.6)
	Two	46 (13.81)
	Three	247 (74.17)
	Four	4 (1.2)
How much do you feel in agreement with the NO-VAX positions expressed during the COVID-19 pandemic?	Totally agree	53 (15.92)
	Partially agree	36 (10.81)
	Neither in agreement nor disagreement	25 (7.51)
	Partially disagree	30 (9.01)
	Totally disagree	189 (56.76)

**Table 3 behavsci-13-00235-t003:** Hierarchical multiple regression analyses on sociodemographic variables (n = 333).

*1.1 Demographics such as age, gender, sexual orientation, and relationship status (R^2^ = 0.045; F = 2.973; p = 0.012)*			
	** *B* **	** *SE* **	** *β* **
Age	−0.016	0.013	−0.073
Gender (Female = 0/Male = 1)	−0.269	0.328	−0.052
Sexual orientation (Heterosexual = 0/LGB+ = 1)	0.772	0.326	0.141 *
Being in a relationship (No = 0/Yes = 1)	−0.628	0.243	−0.144 **
*1.2 Socioeconomic variables (R^2^ = 0.022; F = 1.825; p = 0.124)*			
	** *B* **	** *SE* **	** *β* **
Education level	−0.367	0.175	−0.120 *
Socioeconomic status	−0.137	0.161	−0.048
Residence area (From metropolis to rural area)	−0.160	0.112	−0.081
*1.3 Political and religious orientation (R^2^ = 0.069; F = 4.522; p = 0.002)*			
	** *B* **	** *SE* **	** *β* **
Political Conservativisms (Right winged)	−0.192	0.146	−0.085
Religious Education	0.134	0.142	0.060
Religiousness	0.507	0.135	0.247 ***

* *p* < 0.05; ** *p* < 0.01; *** *p* < 0.001. Social desirability was put as covariate in every regression. Results are not reported for simplicity.

**Table 4 behavsci-13-00235-t004:** Hierarchical multiple regression analyses on monkeypox and COVID-19-related variables (n = 333).

*2.1 Monkeypox attitudes (R^2^ = 0.044; F = 3.773; p = 0.005)*			
	** *B* **	** *SE* **	** *β* **
Monkeypox knowledge	0.013	0.122	0.006
Fake news	0.065	0.135	0.033
Media amplification	−0.371	0.108	−0.223 ***
*2.2 COVID-19 QoL and attitudes (R^2^ = 0.109; F = 6.011; p = 0.000)*			
	** *B* **	** *SE* **	** *β* **
Current perceived QoL	−0.093	0.036	−0.154 **
Worsening QoL attributed to COVID-19 pandemic	0.067	0.029	0.134 *
Had COVID-19 (no/yes)	0.100	0.219	0.026
Number of COVID-19 vaccine doses made	0.571	0.150	0.236 ***
Agreement with No-Vax positions	0.076	0.092	0.051

* *p* < 0.05; ** *p* < 0.01; *** *p* < 0.001. QoL = quality of life. Social desirability was put as covariate in every regression. Results are not reported for simplicity.

**Table 5 behavsci-13-00235-t005:** Hierarchical multiple regression analyses on psychological variables (n = 333).

*3.1 Epistemic trust (R^2^ = 0.105; F = 9.661; p = 0.000)*			
	** *B* **	** *SE* **	** *β* **
Trust	0.039	0.025	0.091
Mistrust	0.019	0.021	0.054
Credulity	0.163	0.040	0.257 ***
*3.2 Need for closure (R^2^ = 0.041; F = 2.327; p = 0.033)*			
	** *B* **	** *SE* **	** *β* **
Order	0.064	0.122	0.036
Predictability	−0.143	0.152	−0.075
Decisiveness	0.089	0.144	0.044
Ambiguity	−0.134	0.172	−0.064
Close-mindedness	0.518	0.159	0.225 ***
*3.3 Psychopathology (R^2^ = 0.112; F = 10.299; p = 0.000)*			
	** *B* **	** *SE* **	** *β* **
Somatization	0.120	0.249	0.034
Depression	0.164	0.206	0.056
Anxiety	0.824	0.225	0.282 ***
*3.4 Mentalized Affectivity (R^2^ = 0.039; F = 3.367; p = 0.010)*			
	** *B* **	** *SE* **	** *β* **
Identifying	0.010	0.027	0.021
Processing	−0.087	0.028	−0.183 **
Expressing	0.052	0.023	0.132 *

* *p* < 0.05; ** *p* < 0.01; *** *p* < 0.001; social desirability was put as covariate in every regression. Results are not reported for simplicity.

**Table 6 behavsci-13-00235-t006:** Hierarchical multiple regression analyses on LGBTQI+ and sexual attitudes (n = 333).

*4.1 LGBTQI+ attitudes (R^2^ = 0.043; F = 3.646; p = 0.006)*			
	** *B* **	** *SE* **	** *β* **
Closeness to LGBTQI+ community	0.265	0.076	0.196 ***
Negative attitudes towards Gay Men	0.082	0.079	0.121
Negative attitudes towards Lesbians	0.001	0.085	0.001
*4.2 Sexual attitudes (R^2^ = 0.009; F = 1.527; p = 0.219)*			
	** *B* **	** *SE* **	** *β* **
Moralism	0.025	0.015	0.09

*** *p* < 0.001; social desirability was put as covariate in every regression. Results are not reported for simplicity.

**Table 7 behavsci-13-00235-t007:** Final hierarchical multiple regression analyses on best predictors emerged (n = 333).

*Final regression Best predictors (R^2^ = 0.324; F = 10.130; p = 0.000)*			
	** *B* **	** *SE* **	** *β* **
Step 1			
Social Desirability	−0.034	0.055	−0.034
Step 2			
Social Desirability (covariate)	0.107	0.051	**0.106 ***
Sexual orientation (Heterosexual = 0/LGB+ = 1)	0.362	0.278	0.068
Being in a relationship (No = 0/Yes = 1)	−0.296	0.217	−0.069
Education level	−0.136	0.151	−0.044
Religiousness	0.441	0.111	**0.195 *****
Media amplifications	−0.315	0.082	**−0.190 *****
Current perceived QoL	−0.061	0.034	−0.100
Worsening QoL attributed to COVID-19 pandemic	0.042	0.027	0.081
Number of COVID-19 vaccine doses made	0.306	0.130	**0.120 ***
Credulity	0.126	0.034	**0.198 *****
Close-mindedness	0.204	0.112	0.089
Anxiety	0.572	0.166	**0.196 *****
Processing	0.018	0.028	0.039
Expressing	0.044	0.020	**0.113 ***
Closeness to LGBTQI+ community	0.090	0.071	0.066

* *p* < 0.05; ** *p* < 0.01; *** *p* < 0.001; QoL = quality of life.

## Data Availability

Data are available on request.

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
