# Peer review of "Who Is Afraid of Monkeypox? Analysis of Psychosocial Factors Associated with the First Reactions of Fear of Monkeypox in the Italian Population"

_behavsci, 2023, doi:10.3390/bs13030235_

Round 1

Reviewer 1 Report

Thanks for the manuscript. Monkeypox is another virus after COVID-19 that has widely drawn attention and even fear. This study explores who is afraid of monkeypox and why. This is of great significance for understanding the differences in attitudes towards monkeypox among different populations. There are still some problems to be solved.

Introduction

The study is not comprehensive enough in its literature review on fear. Fear  has both biological and social explanations, both individual and collective explanations. Currently, the literature review on fear  in this study is far from sufficient.  It is recommended to read the following articles.

Adolphs, R. The biology of fear. Curr. Biol. 2013, 23, R79–R93. 

Darwin, C. The expression of the emotions in man and animals. J. Anthropol. Inst. G.B. Irel. 1873, 2, 444.

Gross, C.T.; Canteras, N.S. The many paths to fear. Nat. Rev. Neurosci. 2012, 13, 651–658. 

Davis, M.; Walker, D.L.; Miles, L.; Grillon, C. Phasic vs sustained fear in rats and humans: Role of the extended amygdala in fear vs anxiety. Neuropsychopharmacology 2010, 35, 105–135. 

Xu, T.; Sattar, U. Conceptualizing COVID-19 and Public Panic with the Moderating Role of Media Use and Uncertainty in China: An Empirical Framework. Healthcare 2020, 8, 249. 

Mobbs, D.; Yu, R.; Rowe, J.B.; Eich, H.; FeldmanHall, O.; Dalgleish, T. Neural activity associated with monitoring the oscillating threat value of a tarantula. Proc. Natl. Acad. Sci. USA 2010, 107, 20582–20586.

Lindsay, S.J.E. Treating children’s fears and phobias—A behavioural approach. Behav. Res. Ther. 1984, 22,.

Additionally, the study should propose a specific analytical framework based on a literature review, which can be integrated with the subsequent path analysis and draw corresponding research hypotheses after logical reasoning.

Methods

From the description of the data, we can see that the proportion of male and female samples is very imbalanced, with males almost twice as many as females. I guess this proportion must be far from the real population ratio of men and women. The author should scientifically evaluate the representativeness of the data and provide relevant supporting evidence.

In the method section, the author gives a detailed introduction to the modules in the questionnaire, but this content may also be included as an appendix. The author does not provide a thorough description of the dependent variables, core independent variables, and control variables in the study. It should be specified what type of variables they are and how they are coded in the analysis. For example, how are 

Religiousness,Media amplifications,Current perceived QoL,Credulity measured in the questionnaire and coded in the analysis.

Results 

The regression models in Tables 4 to 6 lack the corresponding control variables, which is inconsistent with the conventional practice of building models. The author should improve these models by incorporating control variables when constructing different models.

Furthermore, although this study has reached some research conclusions, it seems to lack the corresponding policy implications, which can be supplemented.

Author Response

Reviewer1

Introduction

The study is not comprehensive enough in its literature review on fear. Fear  has both biological and social explanations, both individual and collective explanations. Currently, the literature review on fear  in this study is far from sufficient.  It is recommended to read the following articles.

Adolphs, R. The biology of fear. Curr. Biol. 2013, 23, R79–R93. 

Darwin, C. The expression of the emotions in man and animals. J. Anthropol. Inst. G.B. Irel. 1873, 2, 444.

Gross, C.T.; Canteras, N.S. The many paths to fear. Nat. Rev. Neurosci. 2012, 13, 651–658. 

Davis, M.; Walker, D.L.; Miles, L.; Grillon, C. Phasic vs sustained fear in rats and humans: Role of the extended amygdala in fear vs anxiety. Neuropsychopharmacology 2010, 35, 105–135. 

Xu, T.; Sattar, U. Conceptualizing COVID-19 and Public Panic with the Moderating Role of Media Use and Uncertainty in China: An Empirical Framework. Healthcare 2020, 8, 249. 

Mobbs, D.; Yu, R.; Rowe, J.B.; Eich, H.; FeldmanHall, O.; Dalgleish, T. Neural activity associated with monitoring the oscillating threat value of a tarantula. Proc. Natl. Acad. Sci. USA 2010, 107, 20582–20586.

Lindsay, S.J.E. Treating children’s fears and phobias—A behavioural approach. Behav. Res. Ther. 1984, 22,.

Response: thank you so much for this comment. We have tried to insert a discussion of literature about fear based on reviewer 1 suggestions. We hope that the current version could be clearer and more precise.

Additionally, the study should propose a specific analytical framework based on a literature review, which can be integrated with the subsequent path analysis and draw corresponding research hypotheses after logical reasoning.

Response: thanks to the reviewer also for this point. We have added a brief explanation of the psychodynamic theorical framework used as follows: “The theoretical framework that inspired this study refers to mentalization (Fonagy & Allison, 2014) as a development process that relies on sufficiently good attachment relationships that reflect the extent to which subjective experiences are adequately mirrored by a trusted other to establish secondary representations of one's own subjective experiences (Fonagy, 1998). Research has demonstrated links between attachment security, mentalization and the ability to generate epistemic trust (Fonagy & Allison, 2014). This may lead to the hypothesis that more stable traits (such as mentalization, epistemic trust, closed-mindedness, religiosity) may influence health behavior and social attitudes, especially under conditions of fear triggered by uncertainty and un-known.” As explained later in statistical analysis section “A path analysis model was drawn with a theory-driven mode. Variables referring to more stable traits were hypnotized to have an effect in influencing behavior and attitudes variables.”

Methods

From the description of the data, we can see that the proportion of male and female samples is very imbalanced, with males almost twice as many as females. I guess this proportion must be far from the real population ratio of men and women. The author should scientifically evaluate the representativeness of the data and provide relevant supporting evidence.

Response: thank you for highlighting this point. The group reached is not balanced for 2 main reasons. First the snowball sampling method do not guarantee the possibility to reach a representative sample of a population. The second reason is that cisgender women are more prone to fill psychosocial web surveys of any kind compared to other groups. In any case we have better explained that and in the limitation is reported as follows: “Participants were selected with a “snowball” technique and advertisement on social media; therefore, it is impossible to generalize the results for the Italian population despite the variability of participants involved.”

In the method section, the author gives a detailed introduction to the modules in the questionnaire, but this content may also be included as an appendix. The author does not provide a thorough description of the dependent variables, core independent variables, and control variables in the study. It should be specified what type of variables they are and how they are coded in the analysis. For example, how are Religiousness,Media amplifications,Current perceived QoL, Credulity measured in the questionnaire and coded in the analysis.

Response: Thank you for this comment. We have tried to improve this part. We have included the information required in the text and in the appendix 1 and 2. How variables were treated is described in the statistical analysis section (dependent factor, independent variables and covariate – control variable). We hope that the current version is clearer for the reader.

Results 

The regression models in Tables 4 to 6 lack the corresponding control variables, which is inconsistent with the conventional practice of building models. The author should improve these models by incorporating control variables when constructing different models.

Response: Thank you for this comment. As we have written in the statistical analysis part, social desirability was inserted has control variable and cited also in the tables. For the sake of simplicity and brevity above all, we have not reported the social desirability values of the various steps (tab 3-6) but have only reported them in the final one (tab 7). The same procedure has been used in other articles, even the one we published recently derived from the same database on monkeypox attitudes as a fake news. https://www.frontiersin.org/articles/10.3389/fpsyg.2023.1093763/full

If the reviewer believes it is necessary to report the betas and t's of social desirability, we are happy to include it, although it is a repetition for all step1 in the regressions.

Furthermore, although this study has reached some research conclusions, it seems to lack the corresponding policy implications, which can be supplemented.

Response: Thank you for this comment. We have added some new and more specific policy implications in discussions and conclusions.

Reviewer 2 Report

Who is afraid of monkeypox? Analysis of psychosocial factors associated with the first reactions of fear of monkeypox in the Italian population

Thank you for the opportunity to review this important and relevant contribution about an emerging topic that requires more research and visibility.

Overall, the article is very well written, scientifically sound with important implications to behavioral science. Still, I believe a few changes would improve its chances of being published:

1.     Authors should adhere to the journal’s reference style (unless the Editor agrees with APA style).

2.     Authors should prove more consistent information regarding the measurement’s instruments used, including examples of items.

3.     Lines 108-117 should be included in a “procedures” sub-section.

4.     Lines 118-122 should be included in a “participants” sub-section. Also, please provide more relevant information in the text regarding sociodemographic information since table 1 should be complementary to the text.

5.     Tables 3-7: please provide statistical information about R2 and F.

6.     Please include some implications focusing on health and prevention policies in Italy.

Best wishes.

Author Response

Reviewer 2

Thank you for the opportunity to review this important and relevant contribution about an emerging topic that requires more research and visibility.

Overall, the article is very well written, scientifically sound with important implications to behavioral science. Still, I believe a few changes would improve its chances of being published:

  1. Authors should adhere to the journal’s reference style (unless the Editor agrees with APA style).

Response: thank you so much, we have adapted the reference style

  1. Authors should prove more consistent information regarding the measurement’s instruments used, including examples of items.

Response: thank you for this comment. We have provided more examples and information in the text. Moreover, we have added Appendix 1 and 2 to show some items and how they were coded (as also suggested by the other reviewer). We hope that the present version could be clearer for the reader.

  1. Lines 108-117 should be included in a “procedures” sub-section.

Response: Thank you for this comment. We have modified the manuscript accordingly to reviewer’s suggestions.

  1. Lines 118-122 should be included in a “participants” sub-section. Also, please provide more relevant information in the text regarding sociodemographic information since table 1 should be complementary to the text.

Response: Thank you for this comment. We have modified the manuscript accordingly to reviewer’s suggestions.

  1. Tables 3-7: please provide statistical information about R2and F.

Response: Thank you for this comment. We have added R-squared, F and p on each table as required by the reviewer.

  1. Please include some implications focusing on health and prevention policies in Italy.

Response: Thank you so much for this comment. We have added some implications on the Italian situations for health policies in the discussions and in the conclusions.

Round 2

Reviewer 1 Report

All my concerns have received corresponding responses, and the manuscript has been revised. In addition, the format of the paper needs to be modified according to the corresponding provisions of the journal, for example, the figure and table are generally below the text, not above